# Outcomes after cancer diagnosis in children and adult patients with congenital heart disease in Sweden: a registry-based cohort study

Christina Karazisi ,[1,2] Mikael Dellborg,[1,2,3] Karin Mellgren,[4] Kok Wai Giang,[1,2] Kristofer Skoglund,[1] Peter Eriksson,[1,2,3] Zacharias Mandalenakis[1,2,3]

[1]Department of Molecular and Clinical Medicine, University of Gothenburg, Goteborg, Sweden
[2]Department of Medicine, Geriatrics and Emergency Medicine, Region Västra Götaland, Sahlgrenska University Hospital, Goteborg, Sweden
[3]Adult Congenital Heart Disease Unit, Sahlgrenska University Hospital, Gothenburg, Sweden
[4]Department of Pediatric Oncology, Sahlgrenska University Hospital, Goteborg, Sweden

**Correspondence to**
Dr Christina Karazisi;
christina.karazisi@vgregion.se

## ABSTRACT

**Objective** Patients with congenital heart disease (CHD) have an increased cancer risk. The aim of this study was to determine cancer-related mortality in CHD patients compared with non-CHD controls, compare ages at cancer diagnosis and death, and explore the most fatal cancer diagnoses.

**Design** Registry-based cohort study.

**Setting and participants** CHD patients born between 1970 and 2017 were identified using Swedish Health Registers. Each was matched by birth year and sex with 10 non-CHD controls. Included were those born in Sweden with a cancer diagnosis.

**Results** Cancer developed in 758 out of 67814 CHD patients (1.1%), with 139 deaths (18.3%)—of which 41 deaths occurred in patients with genetic syndromes. Cancer was the cause of death in 71.9% of cases. Across all CHD patients, cancer accounted for 1.8% of deaths. Excluding patients with genetic syndromes and transplant recipients, mortality risk between CHD patients with cancer and controls showed no significant difference (adjusted HR 1.17; 95% CI 0.93 to 1.49). CHD patients had a lower median age at cancer diagnosis—13.0 years (IQR 2.9–30.0) in CHD versus 24.6 years (IQR 8.6–35.1) in controls. Median age at death was 15.1 years (IQR 3.6–30.7) in CHD patients versus 18.5 years (IQR 6.1–32.7) in controls. The top three fatal cancer diagnoses were ill-defined, secondary and unspecified, eye and central nervous system tumours and haematological malignancies.

**Conclusions** Cancer-related deaths constituted 1.8% of all mortalities across all CHD patients. Among CHD patients with cancer, 18.3% died, with cancer being the cause in 71.9% of cases. Although CHD patients have an increased cancer risk, their mortality risk post-diagnosis does not significantly differ from non-CHD patients after adjustements and exclusion of patients with genetic syndromes and transplant recipients. However, CHD patients with genetic syndromes and concurrent cancer appear to be a vulnerable group.

## INTRODUCTION

Nearly 2% of children are born with a congenital heart defect.[1] These defects range from simple and uncomplicated, sometimes

## STRENGTHS AND LIMITATIONS OF THIS STUDY

⇒ The primary strength of this study lies in its nationwide scope; it includes all patients born in Sweden between 1970 and 2017 who has been diagnosed with congenital heart disease.
⇒ The follow-up is from birth, in a country with universal health coverage.
⇒ A major limitation is that it relies solely on register-based data, which excludes clinical details. Consequently, we lacked specific information regarding the cancer treatment administered to patients.
⇒ Moreover, we did not have information on cancer stages.

resolving with age, to extremely complex conditions necessitating multiple surgeries and/or catheter interventions for survival. The prognosis for congenital heart disease (CHD) has improved significantly, with over 97% now expected to reach adulthood.[2]

In previous studies, we have found that the overall cancer risk for CHD patients in Sweden is higher than in matched controls without CHD. Notably, this risk in children and young adults is more than double that of the matched controls.[3 4] Some syndromes are known to have an increased risk of both CHD and cancer, such as Downs and 22q11-deletion syndrome.[5 6] Patients with Fontan circulation have been shown over time to develop hepatic fibrosis or cirrhosis, with risk of hepatocellular carcinoma (HCC).[7 8] An association has also been found between cyanotic CHD and an increased risk of pheochromocytomas and paragangliomas.[9 10]

Several studies have tried to find potential risk factors for cancer in CHD patients. Thymectomy at young age can be a possible risk factor.[4 11] Heart transplantation is possible for CHD patients with end-stage circulatory failure.[12] Studies have shown that

transplant recipients have a more than twofold overall increase of cancer compared with the general population.[13 14] The association between low-dose ionising radiation and cancer risk in CHD patients have also been studied. However, in studies with calculation of radiation doses and adjustment or exclusion of cancer predisposition syndromes and/or transplantation—no association has been found.[15 16] Most likely, the association between CHD and cancer is multifactorial. As age increases, so does the risk of acquired diseases such as cancer. Furthermore, evidence suggests that CHD patients are less likely to participate in screenings for malignancies, including cervical, breast and colon cancer.[17] This raises the question if cancer is found at a later stage in CHD patients, which can affect survival rate. Additionally, given that CHD patients have a higher risk of heart failure compared with controls,[18] this could possibly affect cancer treatment—as this can theoretically put the patient at risk of cardiotoxicity—which also could affect the risk of mortality.

Cancer ranks as the fourth leading cause of death among adult CHD patients, preceded by heart failure, pneumonia and sudden cardiac death.[19] Venkatesh *et al* recently reported on malignancy outcomes in a single-centre cohort comprising 68 adults with CHD and cancer. In this group, 24% of the patients died during a median follow-up period of 5 years, with cancer being the cause of death in 69% of these cases.[20] However, to our knowledge, there has been no studies exploring whether there is an elevated risk of mortality in CHD patients with cancer compared with cancer patients without CHD. Furthermore, there are no studies on long-term outcomes for children with CHD following a cancer diagnosis.

In this study, we aim to investigate the prevalence of cancer-related mortality in paediatric and adult patients with CHD born in Sweden between 1970 and 2017. We also examine whether there is an increased risk of mortality in patients with CHD who also have cancer, compared with a control group of cancer patients without CHD. In addition, we will investigate whether the age at cancer diagnosis and mortality differs between patients with CHD and those without CHD. Furthermore, we will identify the most common fatal cancer diagnoses in this population.

## METHODS

### Study population and design

This nationwide registry-based cohort study used data from the Swedish National Inpatient Register (established in 1964, encompassing all cardiothoracic centres since 1970 and achieving national coverage in 1987), the Swedish National Outpatient Register (complete since 2001) and the Swedish Cause of Death Register (complete since 1968). The International Statistical Classification of Diseases and Related Health Problems (ICD) facilitated the identification of diagnoses. Controls were sourced from the Swedish Total Population Register, which is a register where all Swedish inhabitants are registered in since 1968.

Patients diagnosed with CHD born between 1970 and 2017 were identified for the study. Each CHD patient was paired with 10 controls *without* CHD, matched by sex and birth year. Patients not born in Sweden were excluded. Among 67814 CHD patients and 583709 controls; we identified individuals diagnosed with cancer (758 CHD patients and 3670 controls) for inclusion. The Swedish National Board of Health and Welfare and Statistics Sweden provided case–control identification and facilitated register linkage. In the final dataset, all national registration numbers were replaced with a code, ensuring data anonymisation. Data analysis was conducted from January 2023 to October 2023.

### Definitions

CHD was defined using ICD codes, as outlined in online supplemental table 1. The diagnoses of CHD were classified in accordance with the hierarchical CHD classification

| Table 1 Study population | | |
|---|---|---|
| **Characteristic** | **Congenital heart disease** | **Controls** |
| All patients, No. | 67814 | 583709 |
| Patients with cancer, No. (% of total) | 758 (1.1%) | 3670 (0.6%) |
| Male | 360 (47.5%) | 1663 (45.3%) |
| Female | 398 (52.5%) | 2007 (54.7%) |
| Birth period (patients with cancer), No. (%) | | |
| 1970–1979 | 222 (29.3%) | 1538 (41.9%) |
| 1980–1989 | 158 (20.8%) | 832 (22.7%) |
| 1990–1999 | 169 (22.3%) | 616 (16.8%) |
| 2000–2009 | 130 (17.2%) | 489 (13.3%) |
| 2010–2017 | 79 (10.4%) | 195 (5.3%) |
| Age at cancer diagnosis, No. (%) | | |
| 0–17 years | 451 (59.5%) | 1416 (38.6%) |
| 18+ | 307 (40.5%) | 2254 (61.4%) |
| Lesion group (patients with cancer), No. (%) | | |
| Complex lesions | 120 (15.8%) | |
| Non-complex lesions | 638 (84.2%) | |
| Comorbidities (patients with cancer), No. (%) | | |
| Hypertension | 27 (3.6%) | 44 (1.2%) |
| Diabetes | 12 (1.6%) | 42 (1.1%) |
| Myocardial infarction | 1 (0.1%) | 3 (0.1%) |
| Ischaemic stroke | 8 (1.1%) | 7 (0.2%) |
| Atrial fibrillation | 21 (2.8%) | 10 (0.3%) |
| Heart failure | 46 (6.1%) | 6 (0.2%) |
| Genetic syndromes, n (%) | 159 (21%) | 75 (2.0%) |
| Complex | 42 | 61 |
| Non-complex | 117 | 14 |
| Transplant recipients, n (%) | 36 (4.7%) | 23 (0.6%) |

system initially described by Botto *et al* and later modified by Liu *et al.*[21 22] Patients were divided into two groups: those with complex lesions (lesion group 1–2) and those with non-complex lesions (lesion group 3–6) (online supplemental table 2). Cancer was defined using ICD codes, as specified in online supplemental table 3. Genetic syndromes were defined using ICD codes, detailed in online supplemental table 4. Comorbidities were defined using ICD codes according to online supplemental table 5, prior to the date of cancer diagnosis. Transplant recipients in this article refer to transplant recipients of heart, heart/lung, kidney and liver.

## Statistical analysis

All statistical analyses were performed using R V.3.5.2 (R Foundation for Statistical Computing, Vienna, Austria). Baseline characteristics of patients are described with categorical data shown as numbers and percentages, while continuous data are presented as mean±SD or median and IQR. Descriptive statistics were used to report cancer-related mortality among all patients with CHD, expressed as numbers and percentages. The patients were observed from cancer diagnosis until death or the study's conclusion (31 December 2017).

Mortality incidence rates (IR) were reported per 100 person-years, calculated as the total deaths divided by the cumulative follow-up time of the study population. HRs with 95% CIs were derived using Cox proportional hazard regression models. In these models, the control group served as the reference. The analysis was initially conducted without adjustments, followed by two adjusted models. The first adjusted model accounted for variables such as birth year, age at cancer diagnosis and gender. The second adjusted model included the variables from the first model and additionally adjusted for comorbidities, namely hypertension, diabetes, myocardial infarction, ischaemic stroke, atrial fibrillation and heart failure. A p value of less than 0.05 was considered statistically significant.

## Patient and public involvement

None.

## RESULTS

### Baseline characteristics

We identified 67 814 patients born in Sweden between 1970 and 2017 who had been diagnosed with CHD, along with 583 709 matched controls without CHD who were matched by birth year and sex. Among these, 758 (1.1%) CHD patients and 3670 (0.6%) controls subsequently developed cancer. The characteristics of our study population are shown in table 1. Cancer cases were most commonly found in the eldest birth cohort 1970–1979 in both CHD patients and controls. However, in the CHD patients the proportion of patients in this group was smaller than in the control group (29.3% vs 41.9%) and more evenly distributed between the birth cohorts. The proportion of CHD patients that was diagnosed with first cancer diagnosis at an age below 18 years was larger than in controls (59.5% vs 38.6%). As expected, patients with genetic syndromes were more commonly found in CHD patients than in controls (21% vs 2.0%). Most comorbidities were also more frequently found in CHD patients. The median follow-up duration post-cancer diagnosis was 6.3 years (IQR 1.9–14.5) for CHD patients and 5.3 years (IQR 2.0–11.3) for the control group (online supplemental table 6). The median age at last follow-up was 25.7 years (IQR 13.8–38.5) for CHD patients and 33.5 years (20.5–42.5) for the control group.

### Cancer diagnoses

Figure 1A,B shows the distribution of cancer diagnoses among CHD patients and controls. Among patients who died, the three predominant types diagnosed in those with CHD were haematological (28.5%), ill-defined, secondary and unspecified sites (20.8%) and eye and central nervous system (CNS) tumours (12.6%). In controls, ill-defined, secondary and unspecified sites were the most common

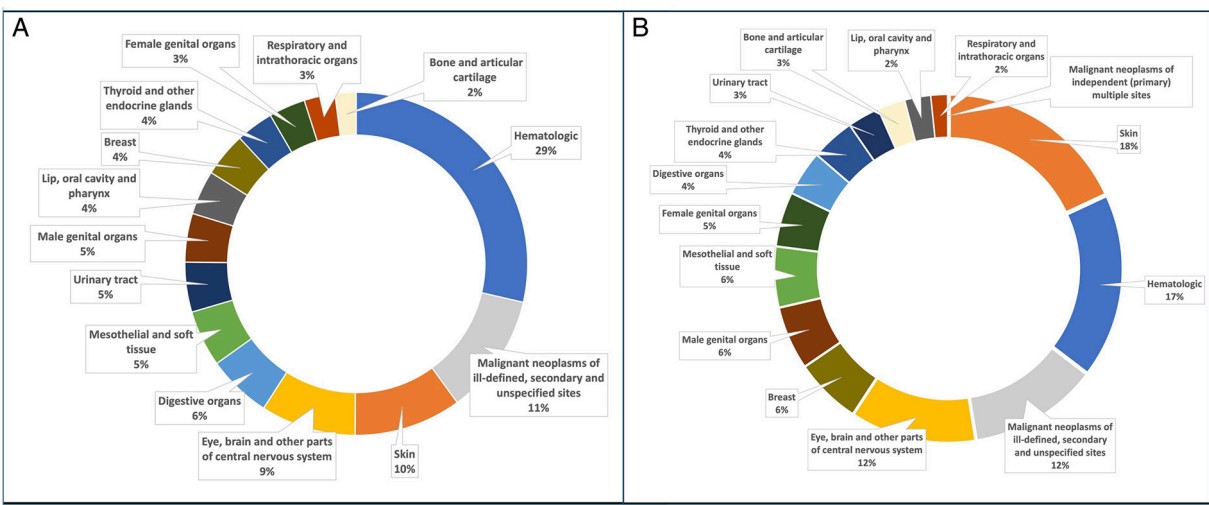

**Figure 1** Cancer diagnoses in (A) congenital heart disease (CHD) patients and (B) controls without CHD.

**Table 2** Deaths

| Patients with syndromes/transplant recipients | Congenital heart disease | | Controls | |
|---|---|---|---|---|
| | Included | Excluded | Included | Excluded |
| Total deaths | | | | |
| In total population, No. (%) | 5689 (0.08%) | 4498 (0.08%) | 4117 (0.007%) | 4000 (0.007%) |
| In patients with cancer (complex/non-complex), No. | 139 (35/104) | 89 (14/75) | 427 | 404 |
| % of deaths in patients with cancer | 18.3% | 15.8% | 11.6% | 11.3% |
| In patients with syndromes (complex/non-complex), No. | 41 (15/26) | | 17 (–) | |
| % of deaths in patients with cancer | 5.4% | | 0.5% | |
| Transplant recipients | <10 | | <10 | |
| % of deaths in patients with cancer | <2% | | <1% | |
| Cancer cause of death* | | | | |
| Total deaths in which cause of death was cancer, No. | 100 | 68 | 370 | 355 |
| Complex lesions, No. (%) | 24 (24%) | 8 (11.8%) | – | – |
| Non-complex lesions, No. (%) | 76 (76%) | 60 (88.2%) | – | – |
| In patients with syndromes, No. (%) (complex/non-complex) | 26 (26%) (11/15) | | 15 (0.4%) (–) | |
| Transplant recipients | <10 | | <5 | |
| Percentage of cancer-related death | | | | |
| In total population deaths | 1.8% | 1.5% | 9.0% | 8.9% |
| In cancer patients | 13.3% | 12.1% | 10.1% | 9.9% |
| In cancer patients deaths | 71.9% | 76.4% | 86.7% | 87.9% |

*Excluding patients with missing causes of death.

(26.0%), followed by eye and CNS tumours (19.3%) and haematological malignancies (15.4%) (online supplemental table 7). When excluding individuals with genetic syndromes and those who had undergone transplants, ill-defined, secondary and unspecified sites emerged as the leading cancer diagnosis in CHD patients, followed by eye and CNS tumours and haematological cancer, a trend consistent in both groups (online supplemental table 8). When calculating case fatality rates of the different cancer types, the only cancer type that differed significantly between CHD patients and the controls was haematological cancers when genetic syndromes and transplant recipients were included—with 21.5% fatality rate in CHD patients versus 14.6% in controls. When excluding patients with genetic syndromes and transplant recipients the difference was no longer significant (13.2% in CHD patients vs 14.0% in controls).

## Mortality after cancer diagnosis

Among the patients studied, 139 patients (18.3%) with concurrent CHD and cancer died during the study period compared with 427 (11.6%) of the control group, who had cancer but not CHD (refer to table 2 and online supplemental table 9 for additional details). Kaplan-Meier survival analysis for patients with cancer, categorised by the presence or absence of CHD, is presented in online supplemental figure 1. Among the 139 deceased patients, 35 (25.2%) had complex lesions, whereas 104 (74.8%) had non-complex lesions. Consequently, the mortality rate was 29.2% among patients with complex lesions and

cancer, compared with 16.3% for patients with cancer having non-complex lesions.

Patients with CHD and cancer exhibited a higher mortality IR compared with controls diagnosed with cancer but without CHD (online supplemental table 10). Notably, the most pronounced differences in mortality IRs were observed in the youngest birth cohort, spanning 2010–2017, in which the IR for CHD patients was 6.13, in contrast to 2.54 for controls.

The initial unadjusted analysis showed a higher mortality risk following cancer diagnosis in patients with CHD compared with controls without CHD (HR 1.55; 95% CI 1.28 to 1.88). This increased risk persisted, although to a lesser extent, after excluding patients with genetic syndromes and transplant recipients (HR 1.35; 95% CI 1.08 to 1.7) (table 3). Subsequent analyses, adjusting for birth period, age at cancer diagnosis and gender in the first model, and additionally for comorbidities in the second model, revealed that the increased risk remained significant (model 1: HR 1.38; 95% CI 1.13 to 1.68; model 2: HR 1.27; 95% CI 1.04 to 1.55). However, when excluding patients with genetic syndromes and transplant recipients, the difference in mortality risk between patients with CHD and the control group was not statistically significant (model 1: HR 1.21; 95% CI 0.96 to 1.53; model 2: HR 1.17; 95% CI 0.93 to 1.49) (table 3, online supplemental table 11).

**Table 3** Mortality after cancer diagnosis

| Mortality HR | HR (95% CI) | |
| --- | --- | --- |
| **All patients** | **Incl. syndromes and tpx** | **Excl. syndromes and tpx** |
| Unadjusted model | **1.55 (1.28 to 1.88)*** | **1.35 (1.08 to 1.70)*** |
| Model 1 (adjusted for birth period, age at cancer diagnosis and gender) | **1.38 (1.13 to 1.67)*** | 1.21 (0.96 to 1.53) |
| Model 2 (adjusted for birth period, age at cancer diagnosis, gender and comorbidities) | **1.27 (1.03 to 1.55)*** | 1.17 (0.93 to 1.49) |

*Numbers in bold denote significance (p<0.05).
tpx, transplant recipients.

## Cause of death

Cancer accounted for 1.8% of mortality cases in the overall CHD cohort, while it represented 72% of deaths among patients with both CHD and cancer (table 2). In fewer than 10 instances among CHD patients with cancer, the cause of death was cardiovascular. Causes of death categorised by complex lesions and non-complex lesions are presented in online supplemental figure 2a,b. Among patients with cancer as the cause of death, 26% (n=26) also had a syndrome diagnosis. All cause of deaths from patients and controls from the year 2017 were missing, because of later registration. Therefore, seven patients who died in 2017 had missing data of cause of death.

## Age at cancer diagnosis

The analysis revealed a notable difference in the age at which cancer was diagnosed between patients with CHD and the control group. Specifically, the median age for cancer diagnosis in patients with CHD was 13.0 years (IQR 2.9–30.0), whereas it was 24.6 years (IQR 8.6–35.1) in the control group (online supplemental table 12). This difference persisted even after the exclusion of patients with genetic syndromes and transplant recipients. However, among patients with cancer who died, the initial observed difference in the age at cancer diagnosis between CHD patients and controls was no longer evident after the exclusion of patients with genetic syndromes and transplant recipients (online supplemental table 12).

## Age at death

The median age at death was lower in patients with CHD and cancer—15.1 years (IQR 3.6–30.7) compared with 18.5 years (IQR 6.1–32.7) in controls with cancer. Excluding patients with genetic syndromes and transplant recipients, the difference in age at death between CHD patients and controls was minimal, with a median age of 19.5 years (IQR 4.7–35.6) for CHD patients versus 18.7 years (IQR 6.1–32.7) for controls (online supplemental table 13). Online supplemental figure 3a,b illustrates the distribution of age at death among patients with CHD and cancer.

## DISCUSSION

This study, to the best of our knowledge, represents the first to explore long-term outcomes following a cancer diagnosis in individuals with CHD, ranging in age from birth to a maximum of 47 years. Our analysis revealed that cancer developed in 758 (1.1%) of the patients with CHD and in 3670 (0.6%) of the controls. In CHD patients, the proportion of patients born in the eldest birth cohort was smaller than in controls: 29.3% of patients with cancer in CHD patients versus 41.9% of patients with cancer in controls. This probably reflects the increased survival in CHD patients; they now more frequently live long enough to develop cancer. This might also have influenced that the proportion of CHD patients diagnosed with cancer below 18 years of age was higher than those above 18 years.

In previous studies, we demonstrated that the most common cancer types in CHD patients were largely similar to those in the control group, with the exclusion of patients with genetic syndromes and transplant recipients.[4] In this study, the most frequent cancer diagnoses for both CHD patients and controls were haematological, ill-defined, secondary and unspecified sites, skin and eye and CNS tumours. The three leading causes of cancer-related fatalities were ill-defined, secondary and unspecified sites, followed by eye and CNS tumours and haematological cancer.

In a nationwide cohort study conducted in Taiwan, Lee *et al* identified an elevated risk of various cancers, particularly haematological, CNS, and head and neck malignancies, in children and adults with CHD.[23] Venkatesh *et al* focusing on adults with CHD who developed cancer, reported the most frequent cancer types as haematological, breast, skin, hepatocellular and urological cancers.[20] Notably, in that study, five out of the seven patients with HCC had previously undergone Fontan palliation, a procedure associated with an increased HCC risk.[24] In our study, we categorised HCC under digestive organ malignancies, which emerged as one of the four most fatal cancers. However, fewer than five CHD patients who succumbed to their illnesses had HCC. This lower incidence in our study might be attributed to the younger age of our patients, considering that HCC risk notably increases over 30 years post-Fontan operation.[24]

During the study, 139 patients with CHD and cancer died, constituting 18.3% of the study population. Among these, cancer was the primary cause of death for 72% of patients with CHD. This finding aligns with the research

by Venkatesh *et al* on adult CHD patients, which reported a 24% mortality rate among patients with cancer over a median follow-up period of 5 years post-cancer diagnosis, with cancer accounting for 69% of these deaths.[20]

In our total CHD cohort, 1.8% of all deaths were attributed to cancer. Previous studies have reported higher frequencies of cancer-related mortality in CHD patients, ranging from 6% to 16%.[19 25 26] However, these studies focused exclusively on adult CHD patients. As age increases, the risk of acquiring other diseases also rises, potentially affecting mortality risk. For instance, in the study reporting 16% cancer-caused mortality, 42% of the patients had heart failure.[26] Conversely, in our CHD and cancer cohort, only 6.1% of patients had heart failure.

A recent study investigating causes of death in infants, children and young adults up to 20 years old with CHD found that the primary cause of mortality in infants was CHD, followed by lung disease and infection. In children, the leading causes were CHD, neurological disease and infection.[27] The absence of cancer in this report likely reflects its relatively low incidence of cancer-related death in paediatric patients compared with adults. Our study, encompassing both children and adult patients, suggests that the generally lower frequency of cancer-related deaths in children might contribute to the reduced frequency of cancer-caused mortality observed in our cohort.

The mortality IRs were notably higher in CHD patients with cancer than in controls with cancer but without CHD. Overall, patients with CHD have an increased mortality risk relative to controls.[28] The highest mortality IR was found in the youngest birth cohort of patients with cancer. However, this is probably due to the low numbers and the fact that this group was the smallest in comparison to the other birth cohorts and generates the smallest cumulative follow-up time as it also is the youngest.

In our analysis of whether patients *with* CHD *and* cancer face a heightened mortality risk compared with controls *without CHD* but *with* cancer, we found an increased risk in patients with CHD when factors such as birth period, age at cancer diagnosis, gender and comorbidities were accounted for (HR 1.27; 95% CI 1.04 to 1.55). However, this increased risk was not significant when excluding patients with genetic syndromes and transplant recipients from the analysis, in the adjusted models.

Mortality rates among patients with CHD have declined over time, especially for those born after 1975.[28] Our study included patients born after 1970 who were diagnosed with cancer. Increased survival rates into adulthood may have altered both the risk of cancer development and the effectiveness of cancer treatment over time. This could explain the lack of a significant difference in survival rates between CHD patients and controls.

In the observed cohort, genetic syndromes accounted for 41 deaths of the total 139 fatalities (29%), of which 26 were attributed to cancer (representing 26% of all cancer-related deaths among CHD patients). Among these 26 cancer-related deaths, 11 occurred in patients with complex lesions (accounting for 46% of cancer-related deaths in this subgroup) and 15 in those with non-complex lesions (constituting 20% of the deaths in this category). Previous studies have shown that children and adolescents with non-Hodgkin's lymphoma with pre-existing conditions such as cancer predisposition syndromes (eg, Down syndrome), immunodeficiencies, genetic disorders (eg, Williams-Beuren syndrome) and other non-classifiable conditions, exhibit lower survival rates compared with their counterparts without such pre-existing conditions.[29] Our findings corroborate the heightened vulnerability of patients with genetic syndromes and concurrent cancer. Case fatality rate of haematological cancer was increased in CHD patients compared with controls when including genetic syndromes and transplant recipients.

In examining the age at cancer diagnosis, it was observed that patients with CHD were diagnosed at a generally younger age than those without CHD. Differences were noted in the age at diagnosis among patients that died, when including those with genetic syndromes and transplant recipients. Recognising the heightened risk of cancer at younger ages is crucial for timely detection and treatment, ultimately enhancing survival outcomes.

## Strengths and limitations

The primary strength of our study lies in its nationwide scope; it includes all patients born in Sweden between 1970 and 2017 who had been diagnosed with CHD. Additionally, the longitudinal tracking of patients from birth enhances the robustness of our findings.

However, our study is not without limitations. It relies solely on register-based data. The Swedish National Inpatient Register were not complete until 1987. Hence, there may be missing data for some patients and controls until the time of full coverage. Furthermore, the registers excludes clinical details. Consequently, we lacked specific information regarding the cancer treatment administered to patients. It is important to note that cancer treatment has evolved over time. Moreover, we did not have access to information on cancer stages other than the diagnosis of ill-defined, secondary and unspecified sites—which not surprisingly, emerged as the most frequent cancer diagnosis in patients who succumbed to their illness.

Another limitation is our focus on cancer mortality rates without considering the diverse types of cancer present in the study group. The heterogeneity in cancer types and their therapies significantly influence survival and mortality rates. Despite these variations, the cancer types in CHD patients and controls exhibited similar frequency patterns, lending a degree of reliability to our findings. Furthermore, since we do not have access to any results of genetic testing, the proportion of genetic disorders might be underestimated. Additionally, it is important to note that Sweden has a universal health coverage system. Consequently, these results may not be generalisable to patients with CHD in countries with different healthcare systems.

## CONCLUSIONS

In conclusion, this study revealed that only 1.8% of deaths among Swedish patients with CHD, with or without cancer, were attributable to cancer. Among those with both CHD and cancer, 18.3% died within a median follow-up duration of 6.3 years (IQR 1.9–14.5). In the CHD patients who died, cancer was identified as the cause of death in 71.9% of cases. Notably, the mortality risk in CHD patients with cancer did not significantly exceed that of controls with cancer in the adjusted models, when patients with genetic syndromes and transplant recipients were excluded from the analysis. These findings indicate that while CHD patients face an increased risk of developing cancer and tend to develop cancer at a younger age, their mortality risk post-cancer diagnosis does not significantly differ from that of patients without CHD, again excluding those with genetic syndromes and transplant recipients.

**Acknowledgements** We thank Phoebe Chi, MD, from Edanz (https://www.edanz.com/ac), for editing a draft of this manuscript.

**Contributors** All authors have contributed significantly to the submitted work. CK, MD, KM, KWG, KS, PE and ZM contributed to the conception and design of the study. CK, ZM and KWG did the experimental design and the analyses. CK and ZM prepared the manuscript and verified the data. CK, MD, KM, KWG, KS, PE and ZM critically reviewed the manuscript and approved the final manuscript. All authors had full access to all the data in the study and had final responsibility for the decision to submit for publication. CK and ZM are responsible for the writing of this manuscript accuracy of the data and accepts full responsibility for the work and/or the conduct of the study, had access to the data, and controlled the decision to publish.

**Funding** The study was funded by the Swedish state under an agreement between the Swedish Government and County Councils (grant number: ALFGBG-955256), the Swedish Research Council (grant number: 2019-00193), the Swedish Heart-Lung Foundation (grant number: 20190724) and the Swedish Childhood Cancer Fund (grant number: SP2017-0012).

**Competing interests** None declared.

**Patient and public involvement** Patients and/or the public were not involved in the design, or conduct, or reporting, or dissemination plans of this research.

**Patient consent for publication** Not applicable.

**Ethics approval** This study was approved by the Gothenburg Regional Research Ethics Board (Gbg 912-16, T 619-18). The tenets of the Declaration of Helsinki were followed, and the need for patient consent was waived as the study used anonymised register-based data. This report follows the Strengthening the Reporting of Observational Studies in Epidemiology reporting guidelines.

**Provenance and peer review** Not commissioned; externally peer reviewed.

**Data availability statement** No data are available. The raw data for our estimates are potentially identifiable, and access to those data is restricted. Researchers wishing to access the individual-level data must apply for permission through a research ethics board and from the primary data owners, the Swedish National Board of Health and Welfare and Statistics Sweden.

**ORCID iD**
Christina Karazisi http://orcid.org/0000-0001-5639-2822

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
