## [Reviewer comments · BMJ Open]

ARTICLE DETAILS

TITLE (PROVISIONAL)	Outcomes after cancer diagnosis in children and adult patients with congenital heart disease in Sweden – a registry-based cohort study
AUTHORS	Karazisi, Christina; Dellborg, Mikael; Mellgren, Karin; Giang, Kok Wai; Skoglund, Kristofer; Eriksson, Peter; mandalenakis, zacharias

VERSION 1 – REVIEW

REVIEWER	Vedel, Cathrine Copenhagen University Hospital
REVIEW RETURNED	02-Feb-2024

GENERAL COMMENTS	Review of " Outcomes after cancer diagnosis in children and adult patients with congenital heart disease" General comments: This nationwide study present data from 1970 to 2017 on cancer mortality in CHD patients compared to controls. The study is well-written and contribute with important findings. I have some smaller concerns I would liked addressed. Introduction: Line 6: Probably a conservative estimate. The incidence of VSDs alone is reported to be between 2-6% in neonatal echocardiography studies. Line 6+13: Write CHD the first time you describe it. And use either "congenital heart defect" or "congenital heart disease" consistently. Four well-defined aims. Methods: P. 7, line 25: You write that the database was "established in 1964, encompassing all cardiothoracic centers since 1970, and achieving national coverage in 1987". Your study includes patients from 1970. I see how you can include CHD cases from the cardiothoracic center, but what about controls? This should be explained more thoroughly. P. 7, line 43: What is the Swedish Total Population Register? Not described above. Suppl. Table 1: You have included many diagnoses as CHDs. Usually diagnoses such as Congenital heart block, Sequestration of the lungs, Secondary hypertension and Congenital phlebectasia are not included. I suggest you do not include them in your study. Suppl. Table 2: You are missing hypoplastic right heart syndrome. I don't think I agree with the grouping of your data. Instead, you should rather consider the possible hypotheses for the association between CHD and cancer. Is it genetic? Or maybe exposure to radiation? Or a combination? Because then your grouping does
--

	not make sense. Then you would rather group those with high association to genetic aberrations (as you do not have information on genetics) or severity. P. 8, line 52: Please compare baseline characteristics. P. 9, line 1-2: “The median follow-up duration was 6.3 years (IQR 1.9–14.5) for the patients with CHD, and 5.3 years (IQR 2.0–11.3) for the controls.” Move to results. Results: P. 9, line 44: Please compare baseline characteristics, as requested above. Table 2: Please provide percentages for all groups and comparisons. You should to a greater extent show what happens, when including/excluding those with genetic disorders. If the association disappears when excluding those with syndromes, it is very apparent extra focus should be paid to those individuals with genetic disorders. Rather than scaring the rest of the patients. Table 3 is interesting and should be a primary finding. Discussion: Somewhere, you need to discuss potential reason for the association. Not only between cancer and CHD, but also why they potentially have a higher risk of dying. It appears it is closely associated with genetic disorders, which needs to be addressed. I also think you should write that you were lacking genetic information, hence, the proportion may have been even higher. Both in the general discussion, but also in limitations. P. 13, l. 28: “The highest mortality IR was found in the youngest birth cohort of patients with cancer.” You should discuss possible reasons for this finding. P. 13, l. 54: Write in past tense about your current study. P. 14: you can remove this paragraph. “In Sweden, excluding malignant melanoma and basal cell carcinoma, skin cancer ranks as the second most common cancer diagnosis. Malignant melanoma is the fifth most common.¹⁷ Our study did not differentiate between skin cancer types; all were categorized under a single group. Skin cancer incidence was higher in controls with cancer but without CHD, compared with CHD patients with cancer. However, it was not a common diagnosis among patients who died. This finding underscores the absence of increased mortality in CHD patients, given the higher incidence of this diagnosis in controls.” P. 14: Move this paragraph up. It is very essential in your findings. “In the observed cohort, syndromes accounted for 41 deaths of the total 139 fatalities (29%), of which 26 were attributed to cancer (representing 26% of all cancer-related deaths among CHD patients). Among these 26 cancer-related deaths, 11 occurred in patients with complex lesions (accounting for 46% of cancer-related deaths in this subgroup) and 15 in those with non-complex lesions (constituting 20% of the deaths in this category). Previous studies have shown that children and adolescents with non-Hodgkin lymphoma, especially those with preexisting conditions such as cancer predisposition syndromes (e.g. Down syndrome), immunodeficiencies, genetic disorders (e.g. Williams-Beuren syndrome), and other nonclassifiable conditions, exhibit lower survival rates compared with their counterparts without such pre-existing conditions.¹⁸ Our findings corroborate the heightened vulnerability of patients with syndromes and concurrent cancer.”
--	---

REVIEWER	Venkatesh, Prashanth
----------	----------------------

	Cedars-Sinai Medical Center
REVIEW RETURNED	21-Feb-2024

GENERAL COMMENTS	The authors need to be congratulated on their manuscript and study tackling the issue of CHD and cancer. Through a well done study, they have answered multiple previously unanswered questions on this very pressing issue on which there are currently inadequate data. The methodology and analysis is sound, and a major strength of this paper is the sample size, as well as the presence of a substantial number of age and sex-matched controls. The registry-based design gives the ability to maintain long-term follow-up, which is crucial to describe in the CHD-cancer cohort. A few comments/ questions:  1. What particular 'transplant' recipients are the author referring to in their analysis and what kind of transplant patients actually were excluded? Did this include heart, non-heart solid organ and stem-cell? Stem-cell transplantation and death from that cause should be counted as cancer mortality. Why were these excluded from the analysis? 2. Could the authors please report case fatality rates of the cancers? i.e. once incident, were some cancers more deadly than others? 3. We have previously reported (Venkatesh et al) the issue of multiple cancers in single patients seen in the adult CHD cohort. Was this observed in your cohort, and if so could the incidence of this be given in the CHD and controls, and also did this affect mortality? 4. Could the authors please provide the median age at last follow-up for the CHD and non-CHD controls (not just the year brackets for birth)? I would also like to see how many of these patients at the time of diagnosis were adults versus pediatric patients – this could be added to supplementary table 12. 5. Could the authors please comment on why the mortality IR was highest in the youngest cohort? I would assume the opposite, especially in light of the data that you have cited from other studies suggesting that cancer-related mortality increases with age.
--

VERSION 1 – AUTHOR RESPONSE

Reviewer #1 Dr. Cathrine Vedel, Copenhagen University Hospital:

This nationwide study present data from 1970 to 2017 on cancer mortality in CHD patients compared to controls. The study is well-written and contribute with important findings. I have some smaller concerns I would liked addressed.

1. Introduction: Line 6: Probably a conservative estimate. The incidence of VSDs alone is reported to be between 2-6% in neonatal echocardiography studies.

We agree that the incidence may be higher, depending on what methods are used in diagnosing CHD. However, in the present article we have not investigated the incidence of

CHD and we are referencing to previous published numbers, which have been 1-2%.

2. Line 6+13: Write CHD the first time you describe it. And use either “congenital heart defect” or “congenital heart disease” consistently.

We have used the abbreviation CHD for congenital heart disease throughout the article. However, when writing that children are born with congenital heart defects, we think it sounds better than writing that children are born with congenital heart disease – therefore the use of both congenital heart defect and disease in this paragraph. However, we have used CHD consistently after this.

3. Methods: P. 7, line 25: You write that the database was “established in 1964, encompassing all cardiothoracic centers since 1970, and achieving national coverage in 1987”. Your study includes patients from 1970. I see how you can include CHD cases from the cardiothoracic center, but what about controls? This should be explained more thoroughly.

For some patients (who were not treated at cardiothoracic centers) and for some controls – there may be missing data until the time of full coverage. We have added this information for clarification in the Limitations-section.

4. P. 7, line 43: What is the Swedish Total Population Register? Not described above.

Thank you for pointing that out! All Swedish inhabitants are registered in the Total Population Register since 1968. The register includes information on birthdate, date of death, emigration, marital status and area of residence. This information has been added to the Methods section – Study population and design.

5. Suppl. Table 1: You have included many diagnoses as CHDs. Usually diagnoses such as Congenital heart block, Sequestration of the lungs, Secondary hypertension and Congenital phlebectasia are not included. I suggest you do not include them in your study.

As you point out, these are not usually included and therefore we have also excluded these from the analysis. We have accidentally submitted a list of ICD codes that we used prior to excluding sequestration of the lungs, secondary hypertension and congenital phlebectasia Supplementary Table 1 have been replaced with the correct list (also ICD8 codes that were used were missing which we have added to all ICD codes).

However, we have included patients with Congenital heart block in the analysis, which is sometimes included in previously published studies. Only 10 patients with congenital heart block with concurrent cancer (who did not also have conotruncal defects, severe conotruncal defects, coarctatio of the aorta, ASD or VSD) were included in this study. Of these, 2 patients died. Thus, the inclusion of congenital heart block or not – should not affect the results.

6. Suppl. Table 2: You are missing hypoplastic right heart syndrome. I don't think I agree with the grouping of your data. Instead, you should rather consider the possible hypotheses for the association between CHD and cancer. Is it genetic? Or maybe exposure to radiation? Or a combination? Because then your grouping does not make sense. Then you would rather group those with high association to genetic aberrations (as you do not have information on genetics) or severity.

Please see our answer to previous comment. Hypoplastic right heart syndrome is included in the analysis and is included in the most current list of diagnoses in updated Supplementary Table 1.

Regarding classifications, there are different classification systems for CHD and this hierarchical classification is commonly used in CHD studies. We believe that this classification is better when comparing CHD patients within a lesion group – because the CHD treatment they have received are more alike in a lesion group between two different individuals born in near birthyears. We agree that it might be better to classify the patients into other categories if one would like to investigate the association between CHD and cancer, however, this was not the intention of the current study.

7. P. 8, line 52: Please compare baseline characteristics.

Thank you for the input! We have added additional information about baseline characteristics in the Results section.

8. P. 9, line 1-2: “The median follow-up duration was 6.3 years (IQR 1.9–14.5) for the patients with CHD, and 5.3 years (IQR 2.0–11.3) for the controls.” Move to results.

Thank you for pointing this out! We have removed this from the Method section, as we accidentally had this sentence both in the Methods and Results section.

9. Results: P. 9, line 44: Please compare baseline characteristics, as requested above.

Thank you for the input! We have added the following: “Cancer cases were most commonly found in the eldest birth cohort 1970-1979 in both CHD patients and controls. However, in the CHD patients the proportion of patients in this group was smaller than in the control group (29.3% vs 41.9%) and more evenly distributed between the birth cohorts. The proportion of CHD patients that was diagnosed with first cancer diagnosis at an age below 18 years was larger than in controls (59.5% vs 38.6%). As expected, patients with syndromes were more commonly found in CHD patients than in controls (21% vs 2.0%). Most comorbidities were also more frequently found in CHD patients.”

Further, we have added following sentence to the discussion section: “In CHD patients, the proportion of patients born in the eldest birth cohort was smaller than in controls: 29.3% of cancer patients in CHD patients vs 41.9% of cancer patients in controls. This probably reflects the increased survival in CHD patients; they now more frequently live long enough to develop cancer. This might also have influenced that the proportion of CHD patients diagnosed with cancer below 18 years of age was higher than those above 18 years.”

10. Table 2: Please provide percentages for all groups and comparisons. You should to a greater extent show what happens, when including/excluding those with genetic disorders. If the association disappears when excluding those with syndromes, it is very apparent extra focus should be paid to those individuals with genetic disorders. Rather than scaring the rest of the patients.

We have updated table 2 with percentages and better comparison of what happens when including/excluding patients with syndromes and transplant recipients.

The fact that the association between CHD and mortality vanishes upon exclusion is of course of great interest! This is something that we are planning to investigate further in upcoming studies! We have added the information that 41 of 139 deaths occurred in patients with syndromes and that CHD patients with syndromes and concurrent cancer appear to be at vulnerable group to emphasize this finding – in the abstract.

11. Table 3 is interesting and should be a primary finding.

We agree that table 3 is a primary finding and we also believe that we do treat these data as a primary finding. If the reviewers think this need further emphasize we would be happy to do so but at this point, we do not understand how this is meant to be done.

12. Discussion: Somewhere, you need to discuss potential reason for the association. Not only between cancer and CHD, but also why they potentially have a higher risk of dying. It appears it is closely associated with genetic disorders, which needs to be addressed. I also think you should write that you were lacking genetic information, hence, the proportion may have been even higher. Both in the general discussion, but also in limitations.

Thank you for the input! We have added the following regarding the association between cancer and CHD in the introduction:

“Some syndromes are known to have an increased risk of both CHD and cancer, such as Downs and 22q11-deletion syndrome. Patients with Fontan circulation have been shown over time to develop hepatic fibrosis or cirrhosis, with risk of hepatocellular carcinoma. An association has also been found between cyanotic CHD and an increased risk of pheochromocytomas and paragangliomas. Several studies have tried to find potential risk factors for cancer in CHD patients. Thymectomy at young age can be a possible risk factor. Heart transplantation is possible for CHD patients with end-stage circulatory failure (12). Studies have shown that transplant recipients have a more than two-fold overall increase of cancer compared to the general population. The association between low-dose ionizing radiation and cancer risk in CHD patients have also been studied. However, in studies with calculation of radiation doses and adjustment or exclusion of cancer predisposition syndromes and/or transplantation – no association has been found. Most likely, the association between CHD and cancer is multifactorial.”

Regarding the potential higher risk of dying, in the paragraph in the introduction regarding cancer screening in CHD patients, following has been added:

“Furthermore, evidence suggest that CHD patients are less likely to participate in screenings for malignancies, including cervical, breast and colon cancer. This raises the question if cancer is found at a later stage in CHD patients, which can affect survival rate. Additionally, given that CHD patients have a higher risk of heart failure compared to controls, this could possibly affect cancer treatment - as this can theoretically put the patient at risk of cardiotoxicity – which also could affect the risk of mortality.”

Regarding genetic information we have added following to the Limitation section:

“Furthermore, since we do not have access to any results of genetic testing, the proportion of genetic disorders might be underestimated.”

13. P. 13, l. 28: “The highest mortality IR was found in the youngest birth cohort of patients with cancer.” You should discuss possible reasons for this finding.

The following have been added to the discussion: “However, this is probably due to the low numbers and the fact that this group was the smallest in comparison to the other birth cohorts and generates the smallest cumulative follow-up time as it also is the youngest”

14. P. 13, l. 54: Write in past tense about your current study.

Good point! We have changed “Our study includes” to “Our study included”.

15. P. 14: you can remove this paragraph. “In Sweden, excluding malignant melanoma and basal cell carcinoma, skin cancer ranks as the second most common cancer diagnosis. Malignant melanoma is the fifth most common.¹⁷ Our study did not differentiate between skin cancer types; all were categorized under a single group. Skin cancer incidence was higher in controls with cancer but without CHD, compared with CHD patients with cancer. However, it was not a common diagnosis among patients who died. This finding underscores the absence of increased mortality in CHD patients, given the higher incidence of this diagnosis in controls.”

Ok, the paragraph has now been removed!

16. P. 14: Move this paragraph up. It is very essential in your findings. “In the observed cohort, syndromes accounted for 41 deaths of the total 139 fatalities (29%), of which 26 were attributed to cancer (representing 26% of all cancer-related deaths among CHD patients). Among these 26 cancer-related deaths, 11 occurred in patients with complex lesions (accounting for 46% of cancer-related deaths in this subgroup) and 15 in those with non-complex lesions (constituting 20% of the deaths in this category). Previous studies have shown that children and adolescents with non-Hodgkin lymphoma, especially those with preexisting conditions such as cancer predisposition syndromes (e.g. Down syndrome), immunodeficiencies, genetic disorders (e.g. Williams-Beuren syndrome), and other nonclassifiable conditions, exhibit lower survival rates compared with their counterparts without such pre-existing conditions.¹⁸ Our findings corroborate the heightened vulnerability of patients with syndromes and concurrent cancer.”

We agree that these findings are very interesting! However, since this was not our pre-decided outcomes, we have chosen to not have this paragraph higher in the Discussion section.

Reviewer #2 Dr. Prashanth Venkatesh, Cedars-Sinai Medical Center:

The authors need to be congratulated on their manuscript and study tackling the issue of CHD and cancer. Through a well done study, they have answered multiple previously unanswered questions on this very pressing issue on which there are currently inadequate data.

The methodology and analysis is sound, and a major strength of this paper is the sample size, as well as the presence of a substantial number of age and sex-matched controls. The registry-based design gives the ability to maintain long-term follow-up, which is crucial to describe in the CHD-cancer cohort.

A few comments/ questions:

1. What particular ‘transplant’ recipients are the author referring to in their analysis and what kind of transplant patients actually were excluded? Did this include heart, non-heart solid organ and stem-cell? Stem-cell transplantation and death from that cause should be counted as cancer mortality. Why were these excluded from the analysis?

Thank you for the comment! We agree that stem-cell transplant recipients should not be excluded – which we have not done. We have clarified in the paper (study design – definitions) that the transplant recipients we have excluded are recipients to transplants of heart, heart/lung, kidney and liver.

2. Could the authors please report case fatality rates of the cancers? i.e. once incident, were some cancers more deadly than others?

Thank you for bringing this up. We have now calculated case fatality-rates and compared these with the fatality rates of the controls. The only cancer type that differed significantly (with statistical certainty) was hematologic cancers when syndromes and transplant recipients were included – with 21.5% fatality rate in CHD patients vs. 14.6% in controls. When excluding patients with syndromes and transplant recipients the difference was no longer significant. This have been added to the results and in the discussion as this further strengthens the vulnerability in patients with syndromes. For some cancer types, even though no significant differences were found, the individuals were too few to proper insure statistical certainty.

3. We have previously reported (Venkatesh et al) the issue of multiple cancers in single patients seen in the adult CHD cohort. Was this observed in your cohort, and if so could the incidence of

this be given in the CHD and controls, and also did this affect mortality?

In our study population, 79.8% of CHD patients had only one cancer diagnosis and 20,2% had ≥ 2 diagnoses. However, this did not differ from the controls, with 79.3% in controls having one cancer diagnosis and 20.7% ≥ 2 diagnoses.

4. Could the authors please provide the median age at last follow-up for the CHD and non-CHD controls (not just the year brackets for birth)? I would also like to see how many of these patients at the time of diagnosis were adults versus pediatric patients – this could be added to supplementary table 12.

The median age at last follow-up was 25.7 years (IQR 13.8-38.5) for CHD patients and 33.5 years (20.5-42.5) for the control group. This has been added to the section Results – Baseline characteristics.

In CHD patients, at the time of cancer diagnosis, the majority of patients were 0-17 years of age. When including patients with syndromes and transplant recipients, 451 (59.5%) of the CHD population, developed cancer at the ages 0-17 years, vs. 1416 (38.6%) in the control group.

This have been added to Table 1 for better clarification for the whole population and in supplementary table 12 the corresponding numbers when incl/excl patients with syndromes and transplant recipients have been added.

(When excluding patients with syndromes and transplant recipients, in CHD patients 299 (53,0%) were 0-17 years old at time of cancer diagnosis vs 1345 (37.6%) in controls)

5. Could the authors please comment on why the mortality IR was highest in the youngest cohort? I would assume the opposite, especially in light of the data that you have cited from other studies suggesting that cancer-related mortality increases with age.

This is probably due to the low numbers and the fact that this group was the smallest in comparison to the other birth cohorts and generates the smallest cumulative follow-up time as it also is the youngest. This have been added to the discussion!

VERSION 2 – REVIEW

REVIEWER	Vedel, Cathrine Copenhagen University Hospital
REVIEW RETURNED	26-Mar-2024

GENERAL COMMENTS	Thank you for the replies and edits. I think you should write genetic syndromes throughout, as "syndromes" may not only reflect those caused by genetic aberrations. Otherwise, congratulations on your nice paper.
---

REVIEWER	Venkatesh, Prashanth Cedars-Sinai Medical Center
REVIEW RETURNED	01-Apr-2024

GENERAL COMMENTS	Edits from prior version have been edited. A sentence-level error: Discussion, 3rd paragraph) should read "...as one of the four most fatal cancers", currently reads "fourth most fatal cancers".
--